# BSA Interaction, Molecular Docking, and Antibacterial Activity of Zinc(II) Complexes Containing the Sterically Demanding Biomimetic N_3_S_2_ Ligand: The Effect of Structure Flexibility

**DOI:** 10.3390/molecules27113543

**Published:** 2022-05-31

**Authors:** Eman Soliman, Mohamed M. Ibrahim, Mohamed E. El-Khouly, Ibrahim El-Mehasseb, Abd El-Motaleb M. Ramadan, Magdy E. Mahfouz, Shaban Y. Shaban, Rudi van Eldik

**Affiliations:** 1Chemistry Department, Faculty of Science, Kafrelsheikh University, Kafrelsheikhheikh 33516, Egypt; emansoliman71020@yahoo.com (E.S.); elmehasseb2@gmail.com (I.E.-M.); ramadanss@hotmail.com (A.E.-M.M.R.); 2Chemistry Department, College of Science, Taif University, Taif 21944, Saudi Arabia; ibrahim652001@yahoo.co; 3Institute of Basic and Applied Sciences, Egypt-Japan University of Science and Technology (E-JUST), Borg El-Arab 21934, Egypt; mohamedelkhouly@yahoo.com; 4Zoology Department, Faculty of Science, Kafrelsheikh University, Kafr El Sheikh 33516, Egypt; mmahfouz4@yahoo.co.uk; 5Department of Chemistry and Pharmacy, University of Erlangen-Nuremberg, 91058 Erlangen, Germany; 6Faculty of Chemistry, Nicolaus Copernicus University, 87-100 Torun, Poland

**Keywords:** zinc thiolate, metallodrug, antibacterial, energy transfer, association affinity

## Abstract

Two zinc(II) complexes, **DBZ** and **DBZH_4_**, that have (ZnN_3_S_2_) cores and differ in the bridging mode of the ligating backbone, effectively bind to BSA. The binding affinity varies as **DBZ** > **DBZH_4_** and depends on the ligand structure. At low concentrations, both complexes exhibit dynamic quenching, whereas at higher concentrations they exhibit mixed (static and dynamic) quenching. The energy transfer mechanism from the BSA singlet excited state to **DBZ** and **DBZH_4_**, is highly likely according to steady-state fluorescence and time-correlated singlet photon counting. Molecular docking was used to support the mode of interaction of the complexes with BSA and showed that **DBZ** had more energy for binding. Furthermore, antibacterial testing revealed that both complexes were active but to a lesser extent than chloramphenicol. In comparison to **DBZH_4_**, **DBZ** has higher antibacterial activity, which is consistent with the binding constants, molecular docking, and particle size of adducts. These findings may have an impact on biomedicine.

## 1. Introduction

Serum albumin is one of the most abundant proteins in the blood and plays an essential function in transporting and disposing of substances in the blood [1]. Most biologically active drugs or molecules have been found to act as carriers through reversible binding to albumin and other serum components [2,3]. Bovine serum albumin (BSA) is the most abundant plasma protein in bovines and consists of 583 amino acid residues in a single chain. Due to its low cost, aqueous solubility, tolerance, and simple preparation, BSA has been widely applied in drug delivery. The structural features of BSA, which consists of a number of amino acid residues, make it capable of binding drugs or bioactive compounds with different physicochemical characteristics [4].

Binding studies of BSA with various molecules were carried out intensively to understand the pharmacokinetics and pharmacodynamics of drugs [5,6,7,8,9]. BSA was selected as a protein model due to its high water solubility [10].

Much interest has been given to zinc thiolate complexes in several metallothioneins and metalloproteins as structural and spectroscopic models for metal binding sites [11,12,13,14,15]. Zinc(II) thiolate imine [Zn(py*^t^*BuN_2_Me_2_S_2_)] (**DBZ**) (py*^t^*BuN_2_Me_2_S_2_^2−^ = 2,6–bis(2–mercapto–3,5–di*-tert*-butylphenyliminoethyl)pyridine) and its saturated analog [Zn(py*^t^*BuN_2_H_4_Me_2_S_2_)] (**DBZH_4_**) (py*^t^*BuN_2_Me_2_H_4_S_2_^2−^ = 2,6–bis(2–mercapto–3,5–di*-tert*-butylphenylaminoethyl)pyridine) were developed in our laboratories and reported elsewhere (Figure 1) [16,17]. We are interested in these kind of complexes because (i) zinc may adopt various coordination geometries; (ii) both ligands are multidentate and have donors of amine N and thiolate S; (iii) the ability of these ligands to regulate the coordination geometry around the zinc center since the py*^t^*BuN_2_Me_2_S_2_ chelate π-acceptor ability stabilizes the square-pyramid geometry; (iv) they can also tune the electron density at the zinc center via both σ donor and π back-bonding effects; (v) these complexes are pentacoordinate zinc centers with a vacant site for biomolecular interactions. These properties can have a significant impact on the reactivity and electrophilicity of the zinc center, and in their biological applications, such as antibacterial activity.

As part of our focus on bioactive materials research [18,19,20,21,22], we report here on the binding of BSA to **DBZ** and **DBZH_4_** complexes using various techniques, e.g., UV–vis absorption and fluorescence, time-correlated single-photon counting, and cyclic voltammetry. The association constants, binding sites, and forces between the two complexes and BSA are reported, discussed, and correlated with the structure. The antibacterial activity of both compounds against Gram-positive (*Escherichia coli* and *Pseudomanse*) and Gram-negative (*Staphylococcus aureus* and *Enterococcus*) bacteria is tested in vitro.

## 2. Results and Discussion

The **DBZ** complex was synthesized as previously reported by template condensation of 2,6-diacetylpyridine with bis(3,5-di-tert-butyl-2-mercaptoaniline)zinc(II) complex in a 1:1 molar ratio [16]. The experimental ^1^H-NMR data along with the DFT (B3LYP/6-31G*) calculated structure [16], showed that the structural index τ [23,24], which represents the relative amount of trigonality (square-pyramid, τ = 0; trigonal-bipyramid, τ = 1; τ = (β –α)/60; where α and β are the two largest angles around the central atom) is 0.42. This reveals that the coordination geometry around zinc in **DBZ** is best described as a five-coordinate zinc-imine structure, and the zinc coordination geometry is between square-pyramidal and trigonal-bipyramidal [16]. **DBZ** is chemically stable in both solution and solid state as its mass spectrum exhibits m/z = 664, which corresponds to the molecular ion peak indicating that the metal is attached to the ligand entity. The zinc amine complex **DBZH_4_** was obtained from **DBZ** by reduction using NaBH_4_ in methanol. The structural index of **DBZH_4_**, τ = 0.33, indicates that the structure could be best described as distorted trigonal-bipyramidal (Figure 2) [16]. The lack of ligand field stabilization accounts for the flexibility of coordination around the zinc ions. This enables dynamic coordination environments of zinc ions, which is important for its catalytic process in enzymes when adopting different coordination numbers in interaction with substrates, for example.

### 2.1. Interaction of BSA with DBZ and DBZH_4_

#### 2.1.1. Emission Studies

BSA’s binding interaction with the zinc complexes **DBZ** and **DBZH_4_** was investigated using various techniques, such as fluorescence, time-correlated single-photon counting, UV–vis absorption, and cyclic voltammetry. BSA fluorescence comes from protein residues (tryptophan, tyrosine, and phenylalanine), and the intrinsic fluorescence is mainly due to tryptophan residues. This protein’s intrinsic fluorescence is highly sensitive to the tryptophan environment, protein conformation transitions, and substrate binding and can give important information about the structures and dynamics [25,26]. Figure 1 shows the fluorescence spectra of the BSA (19 μM) singlet excited state in the absence and presence of **DBZ** and **DBZH_4_** in water. When the excitation wavelength was set at 280 nm, the maximum emission wavelength (λ_em_) of the BSA singlet state was monitored at 340 nm, which may arise mainly from the residue of tryptophan. Based on the maximum fluorescence band, the energy of the BSA singlet excited state was found to be 3.65 eV. As shown in Figure 1, the fluorescence spectra of BSA were strongly reduced by the addition of the two complexes, accompanied by an increase in the emission of the singlet **DBZ** state at 542 nm. For **DBZH_4_**, a clear isosbestic point at 455 nm was observed (Figure 1). The finding that the emission of the BSA singlet state decreases with the formation of new emission bands of the **DBZ** and **DBZH_4_** singlet states, may suggest the occurrence of an energy transfer process from the BSA singlet excited state to the **DBZ** and **DBZH_4_**.

Dynamic quenching occurs when the excited fluorophore is deactivated by interacting with the quencher molecule in solution, whereas static quenching occurs via the creation of a non-fluorescence ground-state complex between the fluorophore and the quencher. The dynamic quenching constant increases with temperature, whereas static quenching constants decrease with increasing temperature [27]. The quenching data can be derived from the Stern–Volmer equation (Equation (1)) [27,28]
*F*_0_/*F* = 1 + *K*_SV_ [Q](1)
where *F*_0_ and *F* represent the fluorescence intensities in the absence and presence of a quencher, *K*_SV_ is the Stern–Volmer constant, *τ*_0_ is the average lifetime of the molecule without quencher, and [Q] is the concentration of the quencher. The *K*_SV_ values are 24.7 × 10^5^ and 0.7 × 10^5^ M^−1^ for **DBZ** and **DBZH_4_**, respectively, indicating that (i) both complexes bind efficiently to BSA compared to that reported in the literature [29]; (ii) **DBZ** binds to BSA nearly 30-fold more than **DBZH_4_** (Table 1). This difference in BSA affinity can be explained on the basis of ligand flexibility and complex structure configuration. The ligand in complex **DBZ** is rigid and adopts the complex arrangement in the square-pyramid and leaves a coordination site empty that is less sterically hindered. On the other hand, the ligand in complex **DBZH_4_** is more versatile and adopts the complex structure with more of a trigonal bipyramidal form, which makes it more difficult for BSA to bind. The *K*_q_ values were derived from Equation (2),
*K*_q_ = *K*_sv_/*τ*_0_(2)
where *τ*_0_ is the average lifetime of BSA in the absence of the quencher. The calculated *K*_q_ values for the formed BSA-complex systems are 5.4 × 10^14^ M^−1^ s^−1^ and 0.15 × 10^14^ M^−1^ s^−1^. The upward curvature of the Stern–Volmer plot was observed at higher concentrations of the **DBZ** and **DBZH_4_** complexes. This anomaly may occur when simultaneous quenching (both dynamic and static) processes take place, and fluorophores may be quenched with the same quencher by both collision and complex formation [30,31]. In this case, Equation (1) can be rewritten as expressed in Equation (3),
*I*_0_/*I* = (1 + *K*_D_ [Q]) (1 + *K*_S_[Q]) = 1 + (*K*_D_ + *K*_S_) [Q] + *K*_D_
*K_S_* [Q]^2^(3)
where *K*_S_ and *K*_D_ are static and dynamic quenching constants, respectively. Equation (3) is second order in [Q] and contributes to an upward curve of *I*_0_/*I* versus [Q] at higher [Q]. The quenching constants shown in Table 1 increase with increasing temperature, which means that the probable quenching mechanism of BSA is a dynamic quenching operation.

#### 2.1.2. Absorption Studies

UV–vis spectroscopy is an important technique for studying interactions between drug molecules and proteins. The UV–vis absorption spectra of BSA (19 μM) in the absence and presence of increasing concentrations of both complexes are reported in Figure 2 to support the quenching mechanism. The low absorption peak at 280 nm showed an increase in intensity in the presence of zinc complexes without any change in wavelength, suggesting that there was no creation of a ground-state complex between BSA and the zinc complexes and that the fluorescence was quenched mainly due to dynamic quenching. The observed absorption peak of BSA at 280 nm with increase in the concentration of both zinc complexes arise from the aromatic amino acids tryptophan and tyrosine.

#### 2.1.3. Time-Resolved Studies

In general, time-resolved estimation is the most reliable approach for differentiating static and dynamic quenching [27]. Time-resolved fluorescence decay of BSA in the absence and presence of **DBZ** and **DBZH_4_** complexes, was obtained by the nanosecond single-photon counting technique at λ_ex_ = 280 and λ_em_ = 340 nm. The time profile of the singlet excited state of BSA control exhibited a single exponential decay with a lifetime of 4.57 ns. With the addition of increasing concentrations of **DBZ** and **DBZH_4_** complexes to solutions of BSA in water, lifetimes of the BSA fluorescence were dramatically reduced as shown in Figure 3 and Figure 4. This further confirms that the dynamic quenching mechanism is responsible for the observed quenching of the tryptophan fluorescence, which indicates the presence of an excited state complex between the zinc complexes and BSA.

#### 2.1.4. Determination of Thermodynamic Parameters

The association constants and number of binding sites between the zinc complexes and BSA were calculated according to Equation (4) [32].
(4)$$\log\frac{F_{0}-F}{F}=\log K_{A}+n\log\left[Q\right]\;$$

A plot of log [(*F*_0_ − *F*)/*F*] vs. log [*Q*] yielded a straight line, whose slope equals *n* and the intercept equals log *K*_A_. *K*_A_ and number of binding sites, *n*, were obtained at various temperatures as shown in Table 2. The values of *n* for **DBZ**–**BSA** and **DBZH_4_**–**BSA** were roughly equal to 1, indicating that there is only one binding site. The binding constants and number of binding sites, *n*, of the interaction between the two complexes and BSA, increase with temperature, which contributes to a greater stability of the BSA complex, and further supports the dynamic quenching mechanism. In general, Δ*G*º reflects the spontaneity of the reaction, whereas Δ*H*º and Δ*S*º are the main quantities for determining the binding force. The thermodynamic parameters of Δ*H*º and Δ*S*º were calculated using Equation (5) [33], and the Gibbs free energy was estimated from Equation (6).
ln*K_A_* = −Δ*H*º/RT + Δ*S*º/R(5)
Δ*G*º = Δ*H*º − TΔ*S*º = −RT ln *K_A_*(6)

As shown in Table 2, both Δ*H*º and TΔ*S*º are positive so that the reaction can be expected to be either entropy or enthalpy driven, and the contribution to the value of Δ*G*º is derived from both TΔ*S*º and Δ*H*º. From a comparison of the values for Δ*H*º and TΔ*S*º, it is clear that the reaction is entropy driven (TΔ*S*º >> Δ*H*º) and involves a large increase in entropy with hydrophobic and electrostatic interactions that play a key role in the binding process. The positive values of Δ*H*º define an endothermic process for which energy is required. The negative values of Δ*G*º suggest that the mechanism of the interaction process is spontaneous at all temperatures [34,35].

#### 2.1.5. Cyclic Voltammetry Studies

Interactions of BSA with the **DBZ** and **DBZH_4_** complexes have also been studied using cyclic voltammetry. Cyclic voltammograms of BSA in water in the presence and absence of zinc complexes were used for a voltammetric study. As shown in Figure 5, the BSA cathodic peak at approximately −0.39 V was moved to a more negative potential (−0.55 V) and (−0.57 V), and the current decreased after the addition of **DBZ** and **DBZH_4_**, respectively. These potential changes suggested that BSA interacts with **DBZ** and **DBZH_4_**, and the interactions are mainly electrostatic. The *k*_R_/*k*_O_ binding constants were determined using Equation (7) [36] and were found to be 0.43 and 0.48 for **DB****Z** and **DBZH_4_**, respectively, and indicate that BSA binds to the oxidized over the reduced form for both complexes,
(7)$$\Delta E=E_{f}-E_{b}=0.059\;\log(\frac{k_{R}}{k_{O}})$$
where *E_b_* and *E_f_* are the formal potentials of the bound and free complex forms, respectively, and *k_R_* and *k_O_* are the corresponding binding constants.

### 2.2. Energy Transfer from BSA to DBZ and DBZH_4_ Complexes

The Förster resonance energy transfer (FRET) mechanism allows the excitation energy to be passed from the donor to the acceptor without photon emission from the former molecular system [37]. According to the FRET principle, the rate of energy transfer depends on the distance between the donor and the acceptor, the orientation of the donor and the acceptor dipoles, and the degree of overlap between the donor emission spectrum and the acceptor absorption spectrum. Energy transfer efficiency (*E*) can be used to evaluate the distance (*r*) between zinc complexes (acceptor) and BSA (donor) in protein using Equation (8),
(8)$$E=1-\frac{F}{F_{0}}=R_{0}^{6}/\left(R_{0}^{6}+r^{6}\right)$$
where *F* and *F*_0_ are the fluorescence intensity of BSA in the presence and absence of the acceptor, respectively, *r* is the distance between the acceptor and donor, *R*_0_ is the critical distance for 50% energy transfer and can be calculated using Equation (9),
(9)$$R_{0}^{6}=8.8\times 10^{-25}K^{2}N^{-4}\emptyset J$$
where the spatial orientation factor of the dipole *K*^2^ = 2/3, the refractive index of the medium (water) *N* = 1.333, and the fluorescence quantum yield of the donor BSA ∅ = 0.15 [10,38]. The spectral overlap integral (*J*), calculated from the overlap of the acceptor UV absorption spectra with the donor fluorescence emission spectra, is given using Equation (10).
(10)$$J=\frac{\int_{0}^{\infty }F\left(\lambda \right)\varepsilon \;\left(\lambda \right)\lambda^{4}d\lambda }{\int_{0}^{\infty }F\left(\lambda \right)d\lambda }\;$$

The overlap of the BSA emission spectra with the absorption spectra of **DBZ** and **DBZH_4_** is shown in Figure 6, and the measured values of *J*, *r*, *R*_0_, and *E* are shown in Table 3. The values of *r* for both complexes are less than 7 nm, suggesting a high probability for non-radiative energy transfer operation [39]. In addition, the distances resulting from this approach are in good agreement with the substrate values in the literature for binding to BSA at site IIA [40]. The slight difference between the values of *r* and *R*_0_ also supported the dynamic quenching mechanism. Based on Equation (8), a plot of E obtained at different concentrations has shown that E gradually increases with increasing concentration of both complexes (Figure 7). The interpretation may be that the number of zinc complex molecules adsorbed on the surface of BSA increases with increasing concentration of the zinc complexes, and as a result, the amount of energy transfer from BSA to the zinc complexes also increases [41].

### 2.3. Antibacterial Activity

All complexes were tested for antibacterial activity against both Gram-positive (*Escherichia coli* and *Pseudo Manse*) and Gram-negative (*Staphylococcus aureus* and *Enterococcus*) bacteria. The antibacterial function of complexes, represented as the diameter of the growth-inhibition region in millimeters, is shown in Figure 8. The minimum inhibitory concentration (MIC) and minimum bactericidal concentration (MBC) of complexes needed to inhibit bacterial growth are presented in Table 4. The MBC is identified by the determination of the lowest concentration of antibacterial agents that reduces the viability of the initial bacterial inoculum by a predetermined reduction of 99.9%. The value agreed on in two or more occasions was adopted as the MIC or MBC strain. Chloramphenicol was used for control purposes. From the inhibition zone, as well as the MIC and MBC data, one can conclude the following: Firstly, both complexes were active, but the activity was less than that for the normal drug (chloramphenicol). Both complexes showed also higher activity compared to the zinc complexes coordinated to thiadiazole ligand as reported by Karcz et al. [42]. The two zinc complexes coordinating one thiadiazole and one acetate, presented in Figure 3, displayed antibacterial effect with an MIC of 0.5 mg/mL against *S. aureus* and of 1.0 mg/mL against *E. coli*.

Secondly, the complex **DBZ** displays greater antibacterial activity relative to complex **DBZH_4_** for Gram-positive and Gram-negative bacteria. Finally, both **DBZ** and **DBZH_4_** complexes displayed higher Gram-positive antimicrobial activity than that against Gram-negative bacteria. The higher antibacterial activity of complex **DBZ** relative to **DBZH_4_** can be accounted for on the basis of the ligand structure and the complex configuration. The ligand in the complex **DBZ** contains double bonds, which reduce the polarity of the zinc ion by means of a π–back bonding between the zinc center and the imine nitrogen. It leads to the delocalization of π–electrons and the partial spread of the positive zinc charge to the donor groups and eventually to the increase in lipophilicity of the complex. Increased lipophilicity improves the penetration of the complex through a lipid membrane and prevents metal binding sites in microorganism enzymes [43]. A second consideration can be the zinc coordination structure that determines the binding of the protein. Complex **DBZ** is located between the square-pyramidal and the trigonal-bipyramidal, which also leaves an empty coordination site open for protein binding. For **DBZH_4_**, the zinc coordination geometry is defined as distorted trigonal-bipyramidal, and in this case, protein binding becomes more difficult [16]. A third reason for the improved antibacterial activity of **DBZ** relative to **DBZH_4_** may be the size of both adducts. To illustrate this, the size distribution of **DBZ—BSA** and **DBZH_4_—BSA** self-assembled in water was calculated using the dynamic light scattering (DLS) technique. As seen in Figure 9, the size range of **DBZ—BSA** was between 90 and 120 nm, with a mean value of 110 nm, which is slightly smaller than that of **DBZH_4_—BSA** (in the range of 100–1000 nm, with a mean size of 339 nm). Such a difference in the size of the self-assembled adduct in water was explained by the existence of strong π–π interactions together with the ionic interactions. The smaller size of **DBZ** increases the cellular absorption of **DBZ** because the antibacterial activity depends heavily on the particle size as stated by Zhang et al. [14]. A fourth factor could be the zeta potential, because Dey et al. [44] stated that the zeta potential can be easily correlated with antibacterial activity. Our data showed that **DBZ** with a −27.8 mV zeta potential exhibited better antibacterial activity compared to **DBZH_4_** (−6.5 mV) (see Figure 10).

Gram-positive bacteria were more prone to both complexes than Gram-negative positive bacteria, while the negative charge on the cell surface of Gram-negative bacteria was greater than that of Gram-positive bacteria [45,46]. The presence of an outer membrane in Gram-negative bacteria can play a key role in our case.

### 2.4. Protein Docking

Molecular docking is crucial for drug discovery and structural molecular biology. It provides a prediction of the binding mode between the complex and a macromolecule such as BSA, in addition providing information regarding the structures of energetically beneficial complexes. BSA functions as a transporter protein and is composed of three identical domains, each of which contains 10 α-helices (Figure 11a) [47]. By using the ICM Pocket Finder property, we could identify 13 different hydrophobic pockets (Table 5). The seven major pockets are distributed over the three domains (Figure 11a) as previously described, which corresponds to long- and medium-chain fatty acid transport [47,48]. The further six smaller pockets located in domain I and III may be responsible for binding small molecules such as drugs (Figure 11b).

After the docking of **DBZ** and **DBZH_4_** against the identified pockets, the docking scores showed that the higher binding affinities were at the smaller pockets, indicating a hydrophobic interaction with the docked compounds. For the major concern with the pocket named HP9, which is located on the domain III at the long helix connecting the two subdomains (Figure 11b), **DBZ** was shown to occupy that pocket more tightly than **DBZH_4_** with binding scores −12.6 and −9.7, respectively. **DBZ** and **DBZH_4_** interact at the binding site with nine common residues (P492, D493, E494, Y496, P498, P536, K537 and A538) and two additional interactive residues, V497 and T539, for **DBZ** (Figure 11b,c). The docking simulations show conformational differences of the docked compounds, which in turn resulted in a geometrical change at the binding site. The reduced **DBZH_4_** looks to be more flexible at the binding site than **DBZ** as it is partially twisted away from the binding site by an angle of 98°, which consequently resulted in loss of the binding with the strong hydrophobic residue V497 (Figure 11d–f). These, however, do not interfere with the predicted hydrophobic binding energy, with a slight change in the van der Waals binding energy (Table 5).

## 3. Experimental Section

### 3.1. Materials

A BSA stock solution (0.4 mM) was prepared by dissolving it in distilled water, and the solution was kept at 4 °C. The zinc(II) complexes **DBZ** and **DBZH_4_** were prepared as described previously [16]. Stock solutions of the zinc(II) complexes (4 mM) were prepared in a mixture of methanol/water in the ratio of 1:9.

### 3.2. Instrumentation

The optical absorptions were performed using a 1.0 cm quartz cell on a JASCO spectrophotometer (V-780). Fluorescence measurements were carried out using the JASCO spectrofluorometer (FP-8300 model). The excitation wavelength was set at 280 nm, and the wavelength of the emission was set to 285–600 nm. Picosecond fluorescence decay profiles were measured using the FluoTime 300 (Pico Quant, Berlin, Germany) by the single-photon counting method. Lifetimes were evaluated using software supplied with the instrument. The time-resolved fluorescence instrumentation was the same as for steady-state experiments. Cyclic voltammograms were carried out on a Gamry setup with a carbon working electrode and a platinum wire as a counter electrode. A silver–silver chloride (Ag/AgCl) electrode was used as a reference electrode. All tests were carried out in oxygen-free solutions, and sodium sulfate was used as an electrolyte support. Dynamic light scattering (DLS) was used to determine the distribution profile of the particle size in the solution.

### 3.3. Antibacterial Activity

The synthesized complexes were tested against both Gram-positive (*Escherichia coli* and *Pseudo Manse*) and Gram-negative (*Staphylococcus aureus* and *Enterococcus*) bacteria for antibacterial activity. The bacteria were incubated by shaking in the thermostat at 30 °C overnight. At this stage, a loop of each culture was placed in 10 mL of 10-fold diluted broth, and the test bacteria cultures containing 10^5^ mL^−1^ cells were used for antimicrobial testing. The ratio of the colony numbers to those without these compounds for the two complexes, **DBZ** and **DBZH_4_**, was used as the surviving cell number; the antimicrobial activity was evaluated using this value. MIC and MBC as recommended by the NCCLS (National Committee for Clinical Laboratory Standards, Albany, NY, USA) [49]. After 24 h of incubation, the MIC was read at 30 °C equivalent to the concentration of the tube without visible growth to measure the effect of decreased antibiotic/antiseptic concentrations over a defined period of time on the inhibition of microbial population growth. These assessments may be quite useful during the R&D phase of the product to determine the appropriate concentrations required in the final product, since the concentration of the drug required to produce the effect is normally several hundreds to thousands of times lower than the concentration found in the completed dosage form. In order to evaluate the MBC, a sample of 100 μL was taken from each tube to an MH agar plate without visible growth and incubated for another 24 h at 30 °C.

### 3.4. Molecular Docking

Molecular docking was performed by the inter-coordinate mechanics (ICM)-based docking using ICM-Pro 3.8 software (MolSoft L.L.C, San Diego, CA, USA). The 3D structures of each compound were generated in order to perform the best-suited docking. The structural models of the BSA binding sites were built using the available crystal structure (2.47 Å; PDB code: 4F5S). At first, the properties involved in the BSA interface were adjusted by the deletion of water molecules, refinement of formal charges, and optimization of hydrogen atoms. Then, we used the ICM pocket finder algorithm to predict the binding cavities and clefts. Upon this, we used each of the generated possible binding pockets as an independent receptor for further optimizing the best score. To improve docking correctly, we applied the SCARE method (scanning and refinement) by replacing a pair of side chains with alanine, which allows the receptor to fit into ligand docking. In addition, we logged on the receptor flexibility for further relaxation to perform a flexible docking. The protein−ligand complexation scores were calculated based on the shape complementarity and electrostatic potential of the ligand and the protein’s binding sites, which were represented by the Gaussian potential.

## 4. Conclusions

Finally, the binding affinities of **DBZ** and **DBZH_4_** to BSA were studied using various spectroscopic techniques, e.g., steady-state absorption and fluorescence, time-correlated singlet photon counting (TCSPC), cyclic voltammetry, and molecular docking. Both complexes tightly bind and quench BSA’s endogenous fluorescence with hydrophobic interactions as the main influences. The binding constants are 24.7 × 10^5^ and 0.7 × 10^5^ M^−1^ for **DBZ** and **DBZH_4_**, respectively, and this indicates that (i) the binding constants are in the optimum range of about 10^4^–10^6^ M^−1^, (ii) complex **DBZ** binds about 35 times stronger than **DBZH_4_**. The much higher affinity of **DBZ** to BSA may arise from the coordination geometry around zinc, which is best described as in between square-pyramidal and trigonal-bipyramidal compared to that of **DBZH_4_** (distorted trigonal-bipyramidal). Using steady-state fluorescence measurements, substantial fluorescence quenching of the BSA singlet excited state at 340 nm by **DBZ** and **DBZH_4_** was observed, followed by the creation of singlet states of both complexes at 450–600 nm, indicating an energy transfer from BSA to **DBZ** and **DBZH_4_**. Using the complementary time-correlated single-photon counting method, the decay time profile of BSA was greatly decreased in the presence of **DBZ** and **DBZH_4_**, confirming the quenching activity of the singlet state of BSA through the energy transfer pathway. According to Förster’s non-radiation energy transfer (FRET) principle, the binding distances (r) between **DBZ-BSA** and **DBZH_4_-BSA** were determined to be 3.21 and 3.67 nm, respectively, as a non-radiation energy transfer mechanism of high probability. Both steady state and time-resolved fluorescence experiments assisted dynamic fluorescence quenching between zinc complexes and BSA. Molecular docking analysis shows conformational differences of both complexes, which in turn results in a geometrical change in the binding site. **DBZH_4_** appears more flexible at the binding site than **DBZ** as it partially twisted away from the binding site by an angle of 98°, which consequently resulted in loss of the binding with the strong hydrophobic residue V497. These factors, however, do not interfere with the predicted hydrophobic binding energy, with a slight change in the van der Waals binding energy. In fact, the antibacterial study against both Gram-positive and Gram-negative bacteria has shown that (i) all complexes were active but lesser than the normal drug (chloramphenicol); (ii) complex **DBZ** displays greater antibacterial activity relative to complex **DBZH_4_** for both Gram-positive and Gram-negative bacteria, which is attributable to the binding constants and particle size of the adducts; (iii) the **DBZ** and **DBZH_4_** complexes displayed higher Gram-positive antimicrobial activity than that against Gram-negative bacteria. Biological applications in metallodrug design can be found in the findings published here.

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
