# Peer review of "BSA Interaction, Molecular Docking, and Antibacterial Activity of Zinc(II) Complexes Containing the Sterically Demanding Biomimetic N3S2 Ligand: The Effect of Structure Flexibility"

_molecules, 2022, doi:10.3390/molecules27113543_

Round 1

Reviewer 1 Report

The subject manuscript is an interesting exploration of the protein binding properties, and potential use as an antibiotic, of two 5-coordinate Zn(II) complexes differing in their ligand oxidation/reduction form from diimine to diamine.  My overall impression is that this work is a good fit for the journal and that with minor revisions, should be accepted for publication.  Specific comments and requests regarding revisions are as follows:

  1. Bottom page 2 and top page 3: The coordination geometries of DBZ and DBZH4 are discussed and the authors narratively describe these as square pyramidal or trigonal bipyramidal. It would be helpful to (1) provide the t5 values for the two complexes as a quantitative expression of these coordination geometries. (Addison, A. W.; Rao, N. T.; Reedijk, J.; van Rijn, J.; Verschoor, G. C. (1984). "Synthesis, structure, and spectroscopic properties of copper(II) compounds containing nitrogen–sulphur donor ligands; the crystal and molecular structure of aqua[1,7-bis(N-methylbenzimidazol-2′-yl)-2,6-dithiaheptane]copper(II) perchlorate". J. Chem. Soc., Dalton Trans. (7): 1349–1356. doi:10.1039/dt9840001349)  (2) A figure should be provided with only the Zn(II) and the 5 coordinated atoms oriented in such a way to demonstrate this geometric difference.  (3) Reference should be made to the crystal structures of the molecules through which these coordination geometries were obtained (and if not by crystal structure, then how they were determined should be explained). (4) Some commentary about solid-state vs. solution structures may also be required to remind the reader that Zn(II) is d10 and flexible in its coordination geometry demands.  (5) Has is been determined experimentally that a 6th ligand binds to the Zn(II) in solution (water)? 

  1. At the top of page 3, the authors state that “DBZ is very stable in both solution and solid state.” This statement is entirely unclear.  Are the authors referring to the ligand itself, or its Zn(II) complex?  What kind of stability is referred to—thermodynamic stability or kinetic stability?  How was this determined?  Are there any quantitative measurements of this stability?  The stability of both complexes in aqueous solution, and/or the buffer used for the BSA binding experiments, is relevant and should be discussed in some way.

  1. Pag 3 line 12 “intestinal” should be something else, probably “intrinsic”

  1. Page 4 last paragraph: Ksv values are reported and it is said they “bind efficiently to BSA compared to that reported in the literature” But then there are no references given and the work “that” is not really defined.  Have these same values for these same complexes already been reported?  Or are they talking about different complexes?  References are needed and clarification of what is being compared to.

  1. Page 9, last paragraph: “the number of BSA molecules adsorbed on the surface of the zinc complexes” Shouldn’t that be reversed?

  1. Page 10: Zn-thiazide complexes are referred to as stronger binding, but no numbers are given and no specific structures are given. These should be added and rationalized.

  1. Page 10-11: The Antibacterial Activity paragraph needs broken into multiple paragraphs, one for each reason why ZnDBZ is stronger than ZnDBZH4.

  1. Page 10-11: Reason 3, the size of ZnDBZ and ZnDBZH4 complexes of BSA as determined by DLS is given as a reason for ZnDBZ stronger antibiotic activity. I think this is a stretch.  I assume the antibiotic activity was for the complex only, not the Complex-BSA adduct.  Size of the BSA adduct may not have much to do with the antibiotic activity.  This seems fairly tentative connection.

  1. Page 10: MIC is discussed by MBC is also tabulated without much discussion at all. The discussion of MBC appears to be located in the Experimental Section.  Some of that Experimental section discussion should be shifted to the text about the antibiotic activity.

Reviewer 2 Report

In this work, a complete analysis of the binding of zinc complex compounds to BSA was carried out. The process has been studied in detail by spectroscopic methods; in addition, analysis by Molecular Docking has been performed. The work is devoted to current issues of biomimetic coordination compounds.

A close reading of the manuscript raised several questions.

1. BSA was used as a transport protein, were similar experiments with HSA performed? If the obtained complexes are supposed to be used as antibacterial drugs, it would be logical to study the interaction with human blood proteins.

2. The data obtained for binding constants with BSA indicate, among other things, a significant proportion of hydrophobic interactions with protein binding centers. Did you determine the maximum number of coordination molecules that bind to the macromolecule?

3. Chloramphenicol, which differs greatly in structure and mechanism of action from the compounds obtained by the authors, was chosen as a reference. It would be more appropriate to give the data, including those for complexes similar in structure, for which the antibacterial activity has been studied.

 After some corrections, the article can be published in Molecules.
